# Effect of Hydrogel Containing *Achyrocline satureioides* (Asteraceae) Extract–Loaded Nanoemulsions on Wound Healing Activity

**DOI:** 10.3390/pharmaceutics14122726

**Published:** 2022-12-06

**Authors:** Lucélia Albarello Balestrin, Patrícia Inês Back, Magno da Silva Marques, Gabriela de Moraes Soares Araújo, Mariana Corrêa Falkembach Carrasco, Matheus Monteiro Batista, Tony Silveira, Jamile Lima Rodrigues, Flávia Nathiely Silveira Fachel, Leticia Scherer Koester, Valquiria Linck Bassani, Ana Paula Horn, Cristiana Lima Dora, Helder Ferreira Teixeira

**Affiliations:** 1Programa de Pós-Graduação em Ciências Farmacêuticas, Universidade Federal do Rio Grande do Sul (UFRGS), Av. Ipiranga 2752, Porto Alegre 90610-000, Rio Grande do Sul, Brazil; 2Programa de Pós-Graduação em Ciências Fisiológicas (PPGCF), Laboratório de Histologia, Instituto de Ciências Biológicas, Universidade Federal do Rio Grande (FURG), Av. Itália, km 8, s/n, Rio Grande 96203-900, Rio Grande do Sul, Brazil; 3Programa de Pós-Graduação em Ciências da Saúde (PPGCS), Laboratório de Nanotecnologia, Faculdade de Medicina, Universidade Federal do Rio Grande (FURG), Rua Visconde de Paranaguá 102, Rio Grande 96203-900, Rio Grande do Sul, Brazil

**Keywords:** *Achyrocline satureioides*, nanoemulsion, hydrogel, healing activity

## Abstract

*Achyrocline satureioides* (Lam.) DC extract–loaded nanoemulsions have demonstrated potential for wound healing, with promising effects on keratinocyte proliferation. We carried out the first in vivo investigation of the wound healing activity of a hydrogel containing *A. satureioides* extract–loaded nanoemulsions. We prepared hydrogels by adding the gelling agent (Carbopol^®^ Ultrez) to extract-loaded nanoemulsions (~250 nm in diameter) obtained by spontaneous emulsification. The final flavonoid content in formulation was close to 1 mg/mL, as estimated by ultra-fast liquid chromatography. Permeation/retention studies using porcine ear skin showed that flavonoids reached deeper layers of pig ear skin when it was damaged (up to 3.2 µg/cm² in the dermis), but did not reach the Franz-type diffusion cell receptor fluid. For healing activity, we performed a dorsal wound model using Wistar rats, evaluating the lesion size, anti-inflammatory markers, oxidative damage, and histology. We found that extract-loaded formulations promoted wound healing by increasing angiogenesis by ~20%, reducing inflammation (tumor necrosis factor α) by ~35%, decreasing lipid damage, and improving the re-epithelialization process in lesions. In addition, there was an increase in the number of blood vessels and hair follicles for wounds treated with the formulation compared with the controls. Our findings indicate that the proposed formulation could be promising in the search for better quality healing and tissue reconstruction.

## 1. Introduction

Wounds are characterized by the rupture in the continuity of the skin or mucous-lining epithelium resulting from illness and shock (physical or thermal) [1]. Small injuries usually heal spontaneously, without the need for interventions. However, larger lesions, such as burns and large breaks in the skin, or even chronic wounds, require quick and efficient attention. In these cases, the injured skin loses its function as a barrier against infections, in addition to allowing other complications, such as hemorrhages [2]. According to the duration, wounds can be characterized as acute (accidents or surgical injuries) or chronic (usually caused by decubitus or burn) [3]. After the injury, the process of skin healing and revitalization begins. Healing occurs in three main stages: The inflammatory phase starts right after tissue injury, followed by cleaning and removal of dead tissues, avoiding infections. Then, there is the repair phase, when the formation of new tissue occurs. Finally, the remodeling phase occurs in which the scar is formed [3,4,5].

The use of extracts, fractions, and/or compounds derived from medicinal plants has been considered for wound healing [6]. *Achyrocline satureioides* (Asteraceae; commonly known as marcela) is a medicinal plant widely used in traditional medicine. There are several biological activities attributed to the extract and flavonoids isolated from this medicinal plant, including anti-inflammatory, antioxidant, and anti-herpetic effects [7,8,9,10]. Additionally, the ability of *A. satureioides* ethanolic extract to increase the proliferation of keratinocytes and fibroblasts has been demonstrated [11]. In vivo, the effect of *A. satureioides* extract dispersed in an ointment demonstrated better collagen renewal in mice, an essential condition for complete wound healing [12].

Considering the potential of *A. satureioides* extract as a wound healing agent and the low aqueous solubility of the flavonoid aglycones quercetin (QCT), luteolin (LUT), and 3-O-methylquercetin (3MQ) [8,11,12], our research group has addressed the feasibility of preparing nanoemulsions containing *A. satureioides* extract [13]. This delivery system features a small droplet size, which may help to overcome the skin barrier, and presents a high drug loading capacity and suitable properties for topical use [14]. We recently demonstrated the potential of nanoemulsions containing the *A. satureioides* extract for topical use in wound healing. They presented promising effects on in vitro cell proliferation and keratinocyte migration, combined with an indication for the absence of cytotoxicity and non-irritating potential [13]. For topical application, the low viscosity of nanoemulsions is a concern, and the design of a semisolid formulation is an important consideration [15,16,17]. The present study is the first to describe the development of semisolid formulations containing *A. satureioides* extract for in vivo topical application in wound healing and to use nanotechnology-based products to improve flavonoid skin retention/permeation, because the small droplet sizes can improve active penetration of lipophilic molecules through the skin, increasing their topical effect.

We aimed to evaluate the effect of a hydrogel containing *A. satureioides* hydroethanolic extract–loaded nanoemulsions (Figure 1) on an experimental in vivo wound healing model (Wistar rats). In the first step, we evaluated the distribution of *A. satureioides* flavonoids from formulations through porcine ear skin using Franz-type diffusion cells. Then, we investigated the ability of the formulation to promote wound healing by evaluating the lesion size, anti-inflammatory markers, oxidative damage, and histology in the rat wound healing model.

## 2. Materials and Methods

### 2.1. Materials

*A. satureioides* was acquired from Centro Pluridisciplinar de Pesquisas Químicas, Biológicas e Agrícolas (CPQBA) da Universidade Estadual de Campinas (São Paulo, Brazil), via a sample deposited in the herbarium with the number 308. Egg yolk lecithin and medium-chain triglycerides were purchased from Lipoid (Ludwigshafen, Germany), polysorbate 80 was acquired from Vetec (Rio de Janeiro, Brazil), and vitamin E was procured from Alpha Química (Cachoeirinha, Brazil). Ultra-fast liquid chromatography (UFLC) required the following reagents: methanol (J.T. Baker, Philipsburg, NJ, USA), acetonitrile (Tedia, Rio de Janeiro, Brazil), and phosphoric acid (Merck, Rio de Janeiro, Brazil). Carbopol^®^ Ultrez 20 was kindly donated by Lubrizol do Brasil Aditivos Ltd.a (São Paulo, Brazil). Albumin for enzyme-linked immunosorbent assay (ELISA) was purchased from Calbiochem (San Diego, CA, USA). Normal goat serum was obtained from R&D Systems (Minneapolis, MN, USA). Protease inhibitor cocktail was purchased from VWR Life Science (Radnor Township, PA, USA). Xylazine and ketamine were purchased from Ceva (Paulínea, Brazil). Ketoprofen was obtained from Merial (Paulínea, Brazil). Finally, thiopental was obtained from Cristália (Itapira, Brazil).

### 2.2. Methods

#### 2.2.1. Preparation of *A. satureioides* Ethanolic Extract

The extract was prepared from the inflorescences of *A. satureioides* by the process of maceration with ethanol 80% (*v*/*v*) for 8 days. The proportion of plant used in relation to the extracting liquid was 7.5:100 (*w*/*v*), and the extract obtained was pressed and filtered according to Balestrin et al. [18].

#### 2.2.2. Preparation of Nanoemulsions and Derived Hydrogels

Nanoemulsions were prepared by spontaneous emulsification according to Bidone et al. [19]. The components of the oil phase were solubilized in ethanol, and this phase was poured over the aqueous phase under constant agitation. The excess solvent was removed by distillation under reduced pressure. The formulations consisted of medium-chain triglycerides, egg yolk lecithin, vitamin E, polysorbate 80, water, and *A. satureioides* extract. The final formulation contained 1% dry residue from the extractive solution.

The hydrogels were prepared by directly adding the gelling agent (Carbopol^®^ Ultrez 20) in the final concentration of 0.15% (HNE_AS_) to the nanoemulsion and the pH adjusted to 7.0 with NaOH. The control formulation (HNE) was prepared without *A. satureioides* extract.

#### 2.2.3. Characterization of Nanoemulsions and Derived Hydrogels

The formulations were characterized by the average droplet size and polydispersity index through photon correlation spectroscopy, after the sample had been diluted in purified water at 25 °C. The zeta potential of the formulations was determined by the electrophoretic mobility of the particles. The analyses were performed after diluting the samples with 1 mM NaCl. The measurements were performed using the Zetasizer Nano-ZS90^®^ (Malvern Instruments, Worcestershire, UK).

#### 2.2.4. Flavonoid Content of *A. satureioides* Ethanolic Extract, Nanoemulsions, and Derived Hydrogels

The levels of the flavonoids QCT, LUT, and 3MQ in the *A. satureioides* ethanolic extract and the formulations were determined using a validated UFLC protocol [20]. The analyses were performed on a Shimadzu Prominence system coupled to a photodiode array (PDA) detector and an automatic injector controlled by the LC-Solution Multi PDA software (Kyoto, Japan). The stationary phase consisted of a Phenomenex Luna C18 column (Phenomenex, Torrance, CA, USA; 100 × 2.0 mm internal diameter; 2.5 µm particle size) protected by an Ultra KrudKatcher in-line pre-column filter (Phenomenex). The mobile phase consisted of methanol and 1% phosphoric acid (48:52) in isocratic mode. The mobile phase flow was 0.3 mL/min and the injection volume was 4 µL. The wavelength was adjusted to 362 nm, and the analysis was performed at 40 °C.

#### 2.2.5. Skin Permeation/Retention

Permeation/retention of QCT, LUT, and 3MQ through porcine ear skin, provided by Cooperativa dos Suinocultores do Caí Superior Ltd.a (São Sebastião do Caí, Brazil), was evaluated using a Franz-type diffusion cell apparatus (DIST, Florianópolis, Brazil). The ear hairs were removed with the aid of scissors and a scalpel, and the ears were cut into circular pieces and frozen on the day the pigs were slaughtered. The pieces were used within 30 days. On the day of the experiment, the pieces were thawed and hydrated with phosphate buffer (pH 7.4) for 30 min. To simulate superficial wounds, tape striping was performed using 15 pieces of tape, and to simulate deeper wounds, the epidermis was removed. Specifically, the skin was left in contact with water for 45 s at 60 °C, and the epidermis was removed with the aid of tweezers. Then, the pieces were placed between the donor and recipient compartment of the Franz diffusion cell. Aliquots of 100 µL of HNE_AS_ were placed on the skin in the donor compartment. The receiving compartment was filled with receiving fluid (acidified water/ethanol 70:30 *v*/*v*). After 8 h, the skin was removed from the apparatus. The excess formulation remaining on the skin was removed with cotton and with the application of a Scotch 3M^®^ tape. Then, the flavonoids were extracted from the skin with methanol for 30 min in an ultrasonic bath. The levels of QCT, LUT, and 3MQ in the porcine ear skin and acceptor fluid were determined with UFLC according to a previously described protocol [16].

#### 2.2.6. In Vivo Wound Healing Assay

##### Animals

Two-month-old male Wistar rats (*Rattus norvegicus*) were used in the wound-healing assay. The animals were acquired from Universidade Federal do Rio Grande do Sul (UFRGS) and maintained in the animal facility of Universidade Federal do Rio Grande (FURG) with *ad libitum* access to water and food. The room temperature and humidity were controlled at 23 ± 2 °C and 55% ± 10%, respectively, and there was a 12-h photoperiod. The animals were acclimated under these conditions for 14 days. The experiment was carried out with four experimental groups each containing 18 animals: a group without treatment (NT), a group treated with hydrogel containing *A. satureioides* extract incorporated in nanoemulsion (HNE_AS_), a group treated with a blank hydrogel (HG), and a group treated with the control hydrogel (HNE).

##### Wound Healing Model

The wound healing model was adapted from Tumen et al. [21]. First, each rat was acclimated for 15 min. Then, the rat received an intraperitoneal injection of xylazine 2% (10 mg/kg) and ketamine 10% (90 mg/kg) for anesthesia. Ketoprofen had been administered subcutaneously for analgesic purposes (5 mg/kg). Then, the dorsal hairs were manually removed, and two dorsal wounds 8 mm in diameter were produced using a sterile surgical punch (Kruuse, Marslev, Denmark). The animals were placed in individual cages with clean wood shaving, under a heat lamp, and containing bags of warm water to maintain their body temperature. Physiological solution was dripped into their eyes to prevent the corneas from drying. All animals were followed up until complete surgical recovery and started to consume food and drink water a few hours after the surgical procedure.

Topical treatments started immediately after the surgery and were performed once daily by administering 15 μL of the formulation. The animals were weighed, and the temperature of the wounds was measured using an infrared thermometer Minipa MT-320 (São Paulo, Brazil) each day throughout the treatment. Further, the dimensions of the wounds were measured with a digital caliper (Kingtools^®^, São Paulo, Brazil) for 12 days. The wound area was calculated to determine wound contraction, according to the equation:$$Wound\;contraction\;\left(\%\right)=\frac{\left(Initial\;wound\;area-Final\;wound\;area\right)}{Initial\;wound\;area}.$$

On days 2 and 7, six animals from each group were euthanized by intraperitoneal thiopental overdose (120 mg/kg), and the healing tissues were removed using a sterile scalpel. In these tissues, analyses were performed to determine the progression of inflammation by determining interleukin-1 (IL-1) and tumor necrosis factor alpha (TNF-α) levels, and myeloperoxidase (MPO) activity. A thiobarbituric acid reactive substances assay (TBARS) was performed to determine the level of reactive oxygen species (ROS). Histological analysis was performed on days 2, 7, and 12 to evaluate epidermal regeneration and angiogenesis.

##### Determination of Inflammation

Determination of MPO Activity

The MPO assay was carried out using the healing tissues removed on day 2 of treatment. The tissues were sprayed with liquid nitrogen and then homogenized with phosphate buffer containing 0.5% hexadecyltrimethylammonium bromide and subjected to three cycles of thermal shocks (bath at 37 °C and liquid nitrogen). The samples were sonicated for 15 s and centrifuged at 10,000 rpm for 15 min at 4 °C. A 25-µL aliquot of the supernatant was mixed with 25 µL of phosphate buffer and 25 µL of 1.6 mM tetramethylbenzidine (in DMSO) in a 96-well plate, which was incubated at 37 °C for 5 min. Then, 100 µL of 0.3 mM H_2_O_2_ was added to each well, and the plate was incubated for 5 min. The absorbance at 650 nm was determined by spectrophotometry after 0, 1, 5, and 10 min.

2.Cytokine Determination

IL-1 and TNF-α levels were evaluated in the scar tissue samples obtained on day 2 of treatment. First, the tissues were powdered with liquid nitrogen in a biopulverizer (BioSpec, Bartlesville, OK, USA). The samples were homogenized using a high-speed homogenizer (Ultraturrax^®^ IKA T-18 Basic, BioVera, Rio de Janeiro, Brazil) and a 1:10 dilution of buffer (*w*/*v*). The diluent buffer was made up of PBS (pH 7.2–7.4), 0.05% Tween 20, 10 mM ethylenediaminetetraacetic acid (EDTA), 0.4 M NaCl, and 2.0% protease inhibitor. After homogenization, the samples were centrifuged at 5000 rpm for 10 min at 4 °C. The supernatant was used to determine the levels of IL-1 and TNF-α with DuoSet^®^ ELISA kits (R&D Systems). The results were normalized in terms of the total protein content according to Lowry et al. [22].

3.Determination of TBARS

The TBARS assay was adapted from Oakes and Kraak [23]. Briefly, the scar tissue samples were powdered as previously described and homogenized with the TBARS diluent (1:10 w:v) containing 8.1% sodium dodecyl sulfate (SDS). The resulting mixture was heated to 95 °C for 30 min. Then, 100 µL of water and 500 µL of *n*-butanol were added. The TBARS diluent is an aqueous solution of 154 µM KCl and 35 µM BHT. The homogenized samples were centrifuged at 5000× *g* for 10 min. Then, for phase separation, a 20-µL aliquot of the supernatant was mixed with 150 µL of 20% acetic acid solution, 150 µL of thiobarbituric acid solution, 50 µL of water, and 20 µL of 8.1% SDS. The fluorescence of the organic phase was evaluated at 553 nm after excitation at 515 nm using a Jenway fluorimeter (model 6380, Dunmow, UK). The results were normalized in terms of the total protein content according to Lowry et al. [22].

4.Determination of the Total Protein Content

The cytokine and TBARS results were normalized in terms of the total protein content. The Lowry method was used to determine the total protein content in the samples [22]. Accordingly, the protein measurement of homogenized tissue was carried out using Folin–Ciocalteu reagent.

5.Histological Analysis

Healing tissues, together with normal skin fragments, were fixed in 10% formalin solution for 12 h and then dehydrated. The tissues were embedded in paraffin (Paraplast^®^, Sigma, St. Louis, MO, USA) and 5-µm-thick slices were obtained. Mallory’s trichrome was used to stain the collagen fibers. The histological slides (three slides/animal) were analyzed using an Olympus BX51 optical microscope (Olympus Co., Tokyo, Japan) at 200× magnification. Blood vessels were counted from the central blade field. Four measurements of epidermal thickness were obtained in different fields and calculated by Image J software.

### 2.3. Statistical Analysis

GraphPad Prism^®^ version 8.0.1 (GraphPad Software, San Diego, CA, USA) was used for statistical analysis. The normality of quantitative data was evaluated by the Shapiro–Wilk test. The homogeneity of variance was evaluated by Levene’s test, O’Brien’s test, and the Brown–Forsythe test. As all data presented normal a distribution and homogeneous variances, they are presented as the mean ± standard deviation. The data were analyzed one-way analysis of variance (ANOVA) followed by the Tukey or Bonferroni (healing activity) post test. *p* < 0.05 was considered significant.

### 2.4. Ethical Aspects

The in vivo healing activity experiments were carried out at FURG, approved by CEUA/FURG (number P072/2016). The animals were under the care of a specialized veterinarian (registration CRMV-RS 15681) during the acclimatization and experimental periods.

## 3. Results and Discussion

### 3.1. Characterization of the Formulations

Prior to evaluating the physiochemical properties of the formulations, we determined the flavonoid content (QCT, LUT, and 3MQ) in the hydroethanolic extract by using UFLC. We found 296.5 ± 12.0 µg/mL of QCT, 153.70 ± 0.14 µg/mL of LUT, and 602.0 ± 18.4 µg/mL of 3MQ, values that are consistent with previous studies that used comparable extraction conditions [19,24]

Table 1 presents the physicochemical properties of the formulations. HNE_AS_ presented an average droplet size close to 250 nm and a polydispersity index of 0.19 (monodispersed formulations). The nanoemulsions exhibited a negative zeta potential related to the presence of negatively charged phospholipids and free fatty acids from egg lecithin. HNE_AS_ had a more negative zeta potential (*p* < 0.05), suggesting adsorption of extract components at the oil/water interface of the nanoemulsions, such as organic acids, as has been reported in the literature [18,19]. The final HNE_AS_ formulation contained 1% (*w*/*v*) *A. satureioides* dry residue. The flavonoid content was close to 1000 µg/mL, with recovery close to 100%.

Overall, the results are consistent with those observed for nanoemulsions prior to thickening with Carbopol^®^ Ultred. Thus, we demonstrated that under the conditions employed in this study, hydrogels containing nanoemulsions with *A. satureioides* extract maintained the evaluated parameters and are consistent with our previous study [13].

### 3.2. Permeation/Retention Studies

Table 2 shows the results of permeation/retention of flavonoids on skin. When skin was impaired with the tape stripping procedure, producing a partial lesion of *stratum corneum*, there was a flavonoid accumulation close to 2.64 µg/cm^2^ (total flavonoids), and a smaller amount in the dermis (0.4 µg/cm^2^). In this condition, there were significantly more flavonoids detected in epidermis and dermis compared with intact skin, as we have reported previously [18]. To gain better insight regarding flavonoid permeation/retention in the context of a wound, we performed an additional experiment after removing the epidermis. We detected more flavonoids in the dermis (up to 3.2 µg/cm^2^, *p* < 0.05). We did not detect flavonoids in the acceptor fluid for either superficial or deep lesions, indicating the preferential accumulation of *A. satureioides* flavonoids in the skin layers [19].

### 3.3. In Vivo Healing Activity

#### 3.3.1. Wound Temperature and Animal Weight

Table 3 shows the weight of animals before the start of treatment until day 12. The weight was not influenced by the different treatments: It increased in all groups. These results indicate that none of the treatments made the animals stop or decrease their food intake. It is worth mentioning that this condition is critical for the healing process, because poor nutrition can negatively affect the body’s defense mechanisms [25].

The wound temperature was evaluated (Table 4) to determine whether the treatments affected inflammation, which can be identified, among other characteristics, by heat. At the beginning, there was a mild state of hypothermia (±33 °C). This hypothermic state may be related to the use of anesthetics due to the direct inhibition of thermoregulation and the decrease in metabolism. After the central nervous system concentration of anesthetics decreases, the tendency is for the temperature to return to normal. This postulate corroborates with our results, because 24 h after surgery, the temperature was around 36 °C [26].

#### 3.3.2. Wound Contraction

To estimate wound contraction, we measured the wound daily during the experiment (from days 0 to 12) with the aid of a digital caliper (Figure 2). We found that for all treatments, there was an increase in the lesion size on day 2 after excisions, which can be considered normal, due to the onset of the inflammatory phase of healing. In general, from day 3 onwards, wound retraction was more noticeable. This finding is consistent with the wound healing process: At this time, the repair phase begins, and new tissue begins to form [5]. On days 1, 2, and 7, the animals treated with HG or HNE presented a tendency toward greater retraction in the wounds area. However, this trend was not significantly different among the treatments (*p* > 0.05). On day 12, all lesions were closed, which indicates that for this parameter, all treatments have the same efficacy.

Pereira et al. [12] compared *Achyrocline satureioides* and *Achyrocline alata* on skin wound healing in mice and found that *A. alata* was more effective because it induced earlier wound closure than *A. satureioides*, although the latter improved collagen renewal compared with *A. alata* [12]. The interesting results that we observed with *A. satureioides* may be related to different phenolic compound content resulting from the differences in the extraction conditions (ethanol content and maceration time), as well as with the formulation used to administer the extract from this medicinal plant [27].

### 3.4. Analysis of Inflammatory Markers and Oxidative Damage

Lipid oxidation generates by-products, mainly malondialdehyde, which react with thiobarbituric acid, producing a pink compound that can be quantified by spectrophotometry. Thus, determination of TBARS is used as an indicator of lipid peroxidation and, consequently, of antioxidant activity. The greater the amount of MDA, the greater the lipoperoxidation. TBARS was quantified at three times in this experiment (on days 2, 7, and 12). Figure 3 shows that the lesions treated with HNE_AS_ presented a tendency toward a lower amount of TBARS independently of the day. These data indicate that these wounds had less lipid damage and, consequently, less oxidative damage. Researchers have shown the antioxidant activity of extracts of *A. satureioides* in different experimental conditions, which has mainly been related to the presence of phenolic compounds [18,28,29,30].

MPO is an important pro-oxidant enzyme with antimicrobial action. This enzyme is released from neutrophils, and its activity is essential for an effective immune response. MPO activity is an indication of the presence of leukocytes and is directly linked to inflammation [31]. Figure 3 shows quantification of MPO activity in the samples analyzed on day 2 of treatment. There was a tendency to increase this enzyme in samples treated with HG and HNE compared with HNE_AS_ and without treatment. Lower MPO activity could indicate a decrease in inflammation.

TNF-α, synthesized mainly by macrophages, monocytes, and neutrophils, is stimulated in the presence of IL-1. TNF-α promotes leukocyte chemotaxis in inflammation as well as fibroblast proliferation, which is essential in the proliferation stage of the healing process [32]. Figure 3 shows the TNF-α levels in the analyzed samples. The lesions treated with HNE present more TNF-α compared with the other samples. This result may indicate a greater exacerbation of inflammation; however, it could also be indicative of fibroblast proliferation, which, as mentioned above, is essential for tissue repair.

Cytokines send various stimulatory, modulatory, or inhibitory signals to different cells of the immune system. Normally, they are found in the body at low concentrations, and their synthesis is increased in response to the presence of an antigen or an unusual situation. IL-1 is an important cytokine produced mainly by monocytes, macrophages, and B lymphocytes and acts as a chemotactic agent in acute inflammation. Similarly, to TNF-α, IL-1 can also stimulate fibroblast proliferation [31]. Furthermore, there were no significant differences among the samples, although there was a slight tendency for elevated IL-1 in wounds treated with HG or HNE (Figure 3).

### 3.5. Histological Analysis

Figure 4 shows the histological analyses, and the values in Table 5 are divided by the treatment and time. Two days after beginning treatment, all images of all groups (HG, HNE, and HNE_AS_) had inflammation, hemorrhage, edema, the presence of remodeled fibroblasts, and collagen deposition. However, at this time, the control group showed 20% less neoangiogenesis relative to the other treatments. At this time, none of the groups presented re-epithelization or the presence of hair follicles. Seven days after starting treatment, all images of all groups presented inflammation, hemorrhage, edema, the presence of remodeled fibroblasts, and collagen deposition. At this time, all images of the HNE and HNE_AS_ groups showed re-epithelization, while the control and HG groups presented 80% and 40% less re-epithelialization, respectively. Twelve days after starting treatment, the control and HG group did not show a reduction in inflammation or hemorrhage. However, lesions treated with HNE and HNE_AS_ showed a 20% reduction in inflammation, in addition to a 60% and 40% reduction, respectively, in the bleeding process. HG showed a 20% reduction in edema formation, while the HNE and HNE_AS_ groups showed a 60% reduction in this process. Further, all treatments presented 100% neoangiogenesis, the presence of remodeled fibroblasts, collagen deposition, and re-epithelialization. Finally, the appearance of hair follicles was observed only in the HNE (20% of samples) and HNE_AS_ (60% of samples) groups.

We measured angiogenesis based on the number of vessels found in each field on days 2, 7, and 12 after starting treatment (Figure 5). After 2 days, the wounds treated with HNE or HNE_AS_ presented around 20 vessels, which represents 4 times the amount found in the lesions that did not receive treatment and around 1.5 times more the wounds treated with pure hydrogel. On days 7 and 12, the wounds treated with HNE_AS_ showed the most vessels (around 30) in the observed fields, while there were 10–20 vessels for the other treatments, but without a significant difference (*p* > 0.05). These data demonstrate that the lesions treated with HNE_AS_ present more blood vessels compared with the other treatments and that this vascularization begins in the first steps of healing, which indicates a potential way that this formulation improves the quality of healing and restores the tissue.

Finally, Figure 6 shows the results of epithelium thickness on day 12 after the injury, because it is at this point that the end of the remodeling phase occurred, when it is finally healed [33]. Although the differences between the groups were not significant, the wounds treated with HNE or HNE_AS_ seemed to have a more expressed epithelium, which indicates a possible improved wound healing quality due to the proposed formulations.

## 4. Conclusions

We successfully produced a hydrogel-thickened nanoemulsion loaded with *A. satureioides* extract. Permeation/retention studies using porcine ear skin showed that flavonoids (QCT, LUT, and 3MQ) from HNE_SA_ could reach deeper skin layers when skin is partially damaged (tape stripping) or after epidermis removal, but did not reach the Franz-type diffusion cell receptor fluid. The results demonstrate that for all treatments administered, the animals did not lose weight, and the temperature of the lesions remained within what was expected (initial drop, followed by normalization over the next hours). The wound contraction data did not reveal significant differences among the groups. Biochemical analyses (TBARS, MPO, IL-1, and TNF-α) revealed a tendency to decrease inflammation and oxidative damage in the HNE_AS_-treated groups. Histological analyses showed that groups treated with HNE and HNE_AS_ showed a decrease in inflammation and hemorrhage, as well as a low formation of edema. We found hair follicles in 60% of the lesions treated with HNE_AS_ and only in 20% for those treated with HNE, while we did not find these appendages in the other groups (ST and HG), suggesting earlier maturation of the restored tissue. There was an increase in angiogenesis in lesions treated with HNE or HNE_AS_, with a marked increase for the latter group. Our results demonstrate that the proposed formulation can be promising when seeking to improve the quality of wound healing and tissue remodeling.

## Figures and Tables

**Figure 1 pharmaceutics-14-02726-f001:**
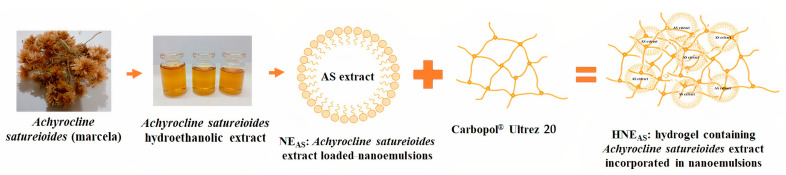
Schematic diagram of hydrogel containing *A. satureioides* hydroethanolic extract–loaded nanoemulsions.

**Figure 2 pharmaceutics-14-02726-f002:**
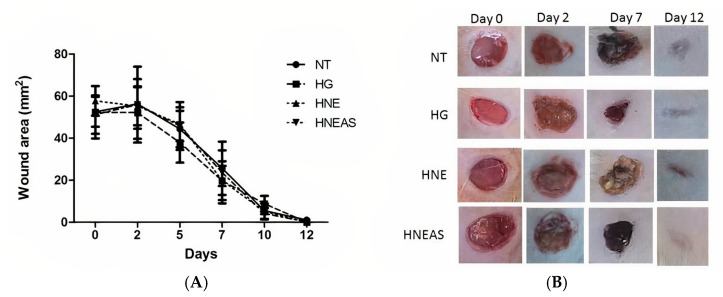
Dorsal wound area in adults male Wistar rats (*Rattus norvegicus*) on days 1, 2, 7, or 12 after injury and daily local application of treatments based in the plant marcela (*Achyrocline satureioides*) extracts and controls. (**A**) Graph comparing the wound retraction data among the groups and over time. Black bar: without treatment; medium gray bar: hydrogel treatment only; dark gray bar: hydrogel treatment containing blank nanoemulsion; and light gray bar: hydrogel treatment containing AS extract incorporated in nanoemulsions. (**B**) Close-up photographs of the wound retraction evolution. NT: no treatment; HG: hydrogel treatment only; HNB: hydrogel treatment containing blank nanoemulsion; and HNE_AS_: hydrogel treatment containing extract of *A. satureioides* incorporated in nanoemulsions. Analysis of variance followed by the Tukey test, *p* > 0.05, *n* = 12.

**Figure 3 pharmaceutics-14-02726-f003:**
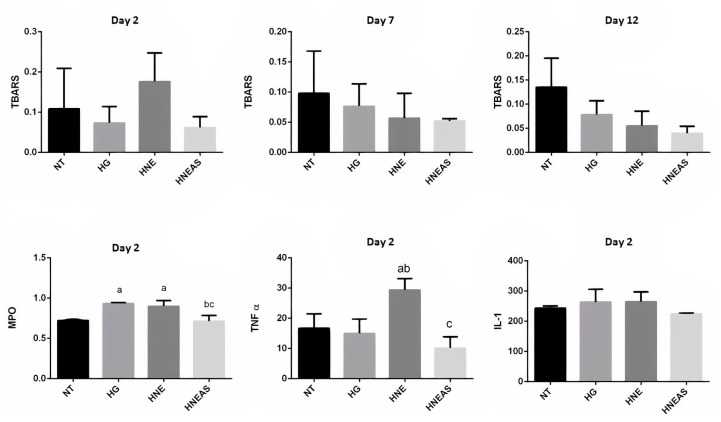
Quantification of thiobarbituric acid reactive substances (TBARS) (nmol/g of protein) of dorsal wounds in adults male Wistar rats (*Rattus norvegicus*) on days 2, 7, or 12 after injury and daily local application of treatments based in the plant marcela (Achyrocline satureioides) extracts and controls. MPO (OD/mg of tissue): myelopeoxidase enzyme; TNF-α (pg/mg of protein): factor of tumor necrosis α; IL-1(pg/mg of protein): interleukin 1. NT: without treatment; HG: hydrogel; HNE: hydrogel containing blank nanoemulsion; and HNE_AS_: hydrogel containing hydrogel incorporated in nanoemulsion. Analysis of variance followed by the Tukey test, a: there is statistical difference with NT (*p* < 0.05); b: there is statistical difference with HG (*p* < 0,05) and c: there is statistical difference with HNE (*p* < 0.05); *n* = 6.

**Figure 4 pharmaceutics-14-02726-f004:**
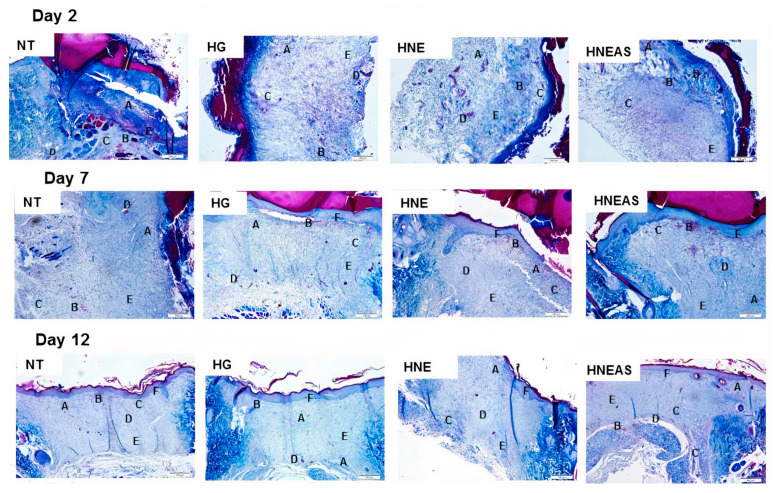
Histological images of dorsal wounds in adults male Wistar rats (*Rattus norvegicus*) obtained 2, 7, or 12 days after injury and daily local application of treatments based in the plant marcela (*Achyrocline satureioides*) extracts and controls. ST: without treatment; Hg: hydrogel treatment; HNE: treatment with blank nanoemulsion; and HNE_AS_: treatment with hydrogel incorporating *A. satureioides* extract. (A) inflammatory process; (B) hemorrhage; (C) edema; (D) neoangiogenesis; (E) collagen; and (F) re-epithelialization. Mallory’s trichrome staining. Magnification: 200×, scale 100 µm.

**Figure 5 pharmaceutics-14-02726-f005:**
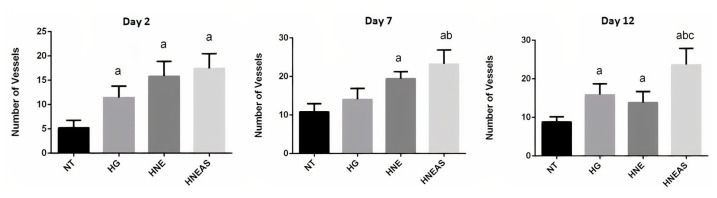
Number of vessels in the skin of dorsal wounds in adults male Wistar rats (*Rattus norvegicus*) on days 2, 7, or 12 after injury and daily local application of treatments based in the plant marcela (*Achyrocline satureioides*) extracts and controls. ST: without treatment; HG: hydrogel treatment; HNE: hydrogel treatment containing blank nanoemulsion; and HNEAS: hydrogel containing *A. satureioides* extract incorporated in nanoemulsion. Analysis of variance followed by the Tukey test, a: there is statistical difference with NT (*p* < 0.05); b: there is statistical difference with HG (*p* < 0.05) and c: there is statistical difference with HNE (*p* < 0.05); *n* = 5.

**Figure 6 pharmaceutics-14-02726-f006:**
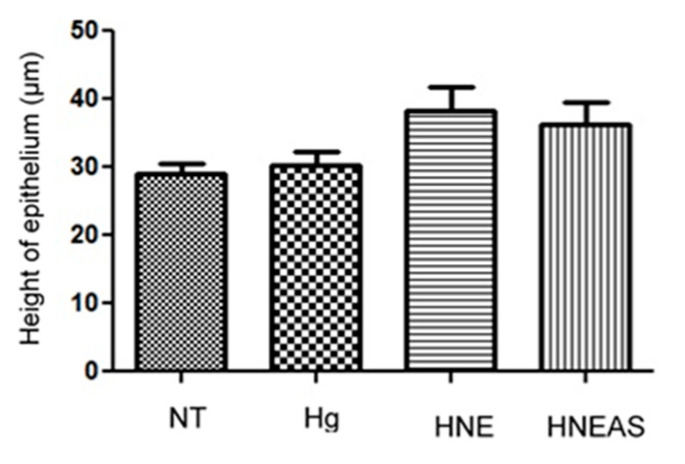
Epithelial estratification of dorsal wounds in adults male Wistar rats (*Rattus norvegicus*) 12 days after injury and daily local application of treatments based in the plant marcela (*Achyrocline satureioides*) extracts and controls. ST: without treatment; HG: hydrogel treatment; HNE: hydrogel treatment containing blank nanoemulsion; and HNEAS: hydrogel containing *A. satureioides* extract incorporated in nanoemulsion. Analysis of variance followed by the Tukey test, *p* > 0.05, *n* = 5.

**Table 1 pharmaceutics-14-02726-t001:** Characterization of formulations applied in evaluations of wound healing from adults male Wistar rats (*Rattus norvegicus*) showed in the present study.

	Droplet Size (nm)	PI	ζ-Potential (mV)	Flavonoids Content (µg/mL)
HNE_AS_	250 ± 3.9	0.19 ± 0.09	−48.0 ± 2.6 ^a^	1086.6 ± 1.9
HNE	210 ± 2.1	0.17 ± 0.01	−27.7 ± 4.0	-

HNE_AS_: hydrogel containing *Achyrocline satureioides* extract incorporated in nanoemulsions; HNE: hydrogel containing blank nanoemulsion. The hydrogels were obtained by adding 0.15% of Carbopol^®^ Ultrez 20 directly to the nonemulsion. PI: Polydispersity index. Deviations are expressed as relative standard deviation (%). Analysis of variance followed by the Tukey test, ^a^ statistical difference with HNE (*p* < 0.05)*, n* = 3.

**Table 2 pharmaceutics-14-02726-t002:** Permeation/retention of flavonoids from HNE_AS_ in the different skin layers of porcine ear skin using Franz-type diffusion cells.

	Tape Stripping			Without Epidermis		
	QCT	LUT	3MQ	QCT	LUT	3MQ
Epidermis (µg/cm^2^)	0.62 ± 4.10	0.42 ± 5.70	1.60 ± 3.9	-	-	-
Dermis (µg/cm^2^)	0.12 ± 9.30	0.09 ± 12.0	0.19 ± 6.70	0.8 ± 8.3 ^a^	0.5 ± 11.0 ^a^	1.9 ± 9.7 ^a^
Fluid (µg/mL)	<LOQ	<LOQ	<LOQ	<LOQ	<LOQ	<LOQ

Where: QCT: Quercetin; LUT: Luteolin; and 3MQ: 3-O-methylquercetin; LOQ: Limit of quantification; Deviations are expressed as relative standard deviation (%). Analysis of variance followed by the Tukey test, ^a^ statistical difference in Dermis between Without Epidermis and Tape Stripping (*p* < 0.05)*, n* = 5.

**Table 3 pharmaceutics-14-02726-t003:** Evolution of weight of adults male Wistar rats (*Rattus norvegicus*) on days 0, 1, 2, 7, or 12 of dorsal wound healing evaluation after injury and daily local application of treatments based in marcela (*Achyrocline satureioides*) extracts and controls.

	Day 0	Day 1	Day 2	Day 7	Day 12
NT	394.8 ± 23.7	394.4 ± 25.4	393 ± 25.2	403.4 ± 27.6	422.6 ± 29.3
HG	399.5 ± 13.0	396.0 ± 13.4	393.3 ± 14.9	405.3 ± 17.1	422.6 ± 20.9
HNE	397.8 ± 24.1	389.6 ± 23.6	387 ± 22.1	389.6 ± 30.8	406.8 ± 36.3
HNE_AS_	397.6 ± 24.6	392.5 ± 28.7	390.1 ± 32.8	404.5 ± 29.1	422.8 ± 29.6

HG: animals treated with blank hydrogel; HNE: animals treated with blank nanoemulsion incorporated in hydrogel; and HNE_AS_: animals treated with AS extract incorporated in nanoemulsions; NT (No treatment). Deviations are expressed as relative standard deviation (%). Analysis of variance followed by the Tukey test, *p* > 0.05, *n* = 6.

**Table 4 pharmaceutics-14-02726-t004:** Evolution of dorsal wounds temperature of adults male Wistar rats (*Rattus norvegicus*) on days 0, 1, 2, 7, or 12 after injury and daily local application of treatments based in marcela (*Achyrocline satureioides*) extracts and controls.

	Day 0	Day 1	Day 2	Day 7	Day 12
NT	32.8 ± 3.89	35.2 ± 5.11	34.0 ± 5.55	35.7 ± 4.52 ^a^	35.6 ± 5.42 ^a^
HG	33.4 ± 2.20	34.8 ± 4.80	33.1 ± 9.17	36.4 ± 2.95 ^a,b^	35.2 ± 5.29
HNE	33.7 ± 2.40	33.9 ± 7.07	34.7 ± 2.86	34.0 ± 4.22	34.7 ± 4.83
HNE_AS_	33.7 ± 2.31	35.2 ± 2.14	34.1 ± 4.83	35.1 ± 3.52	35.7 ± 3.13 ^a^

HG: animals treated with blank hydrogel; HNE: animals treated with blank nanoemulsion incorporated in hydrogel and HNE_AS_: animals treated with AS extract incorporated in nanoemulsions; NT: No treatment. Deviations are expressed as relative standard deviation (%). Analysis of variance followed by the Tukey test, ^a^ statistical difference with Day 0 (*p* < 0.05), ^b^ statistical difference with Day 2 (*p* < 0.05), *n* = 12.

**Table 5 pharmaceutics-14-02726-t005:** Histological data of dorsal wounds in adults male Wistar rats (*Rattus norvegicus*) obtained on days 2, 7, or 12 after local application of treatments based on *Achyrocline satureioides* extracts and controls.

	Day 2				Day 7				Day 12			
	NT	HG	HNE	HNE_AS_	NT	HG	HNE	HNE_AS_	NT	HG	HNE	HNE_AS_
Inflammation (%)	5(5)	5 (5)	5 (5)	5 (5)	5 (5)	5 (5)	5 (5)	5 (5)	5 (5)	5 (5)	4 (5)	4 (5)
	100	100	100	100	100	100	100	100	100	100	80	80
Bleeding (%)	5(5)	5 (5)	5 (5)	5 (5)	5 (5)	5 (5)	5 (5)	5 (5)	5 (5)	5 (5)	2 (5)	3 (5)
	100	100	100	100	100	100	100	100	100	100	40	60
Edema (%)	5(5)	5 (5)	5 (5)	5 (5)	5 (5)	5 (5)	5 (5)	5 (5)	5 (5)	5 (5)	2 (5)	2 (5)
	100	100	100	100	100	100	100	100	100	100	40	40
Neoangiogenesis (%)	4(5)	5 (5)	5 (5)	5 (5)	4 (5)	5 (5)	5 (5)	5 (5)	4 (5)	5 (5)	5 (5)	5 (5)
	80	100	100	100	80	100	100	100	80	100	100	100
Remodeled fibroblasts (%)	5(5)	5 (5)	5 (5)	5 (5)	5 (5)	5 (5)	5 (5)	5 (5)	5 (5)	5 (5)	5 (5)	5 (5)
	100	100	100	100	100	100	100	100	100	100	100	100
Collagen deposition (%)	5(5)	5 (5)	5 (5)	5 (5)	5 (5)	5 (5)	5 (5)	5 (5)	5 (5)	5 (5)	5 (5)	5 (5)
	100	100	100	100	100	100	100	100	100	100	100	100
Re-epitalization (%)	0(0)	0 (0)	0 (0)	0 (0)	1 (5)	3 (0)	5 (5)	5 (5)	5 (5	5 (5)	5 (5)	5 (5)
	0	0	0	0	20	60	100	100	100	100	100	100
Hair follicle (%)	0(0)	0 (0)	0 (0)	0 (0)	0 (0)	0 (0)	0 (0)	0 (0)	0 (0)	0 (0)	1 (5)	3 (5)
	0	0	0	0	0	0	0	0	0	0	20	60

NT: no treatment; HG: hydrogel; HNE: hydrogel containing blank nanoemulsion; HNE_AS:_ hydrogel conteining *A. satureioides* extract–loaded nanoemulsion. Qualitative histological analysis was performed on slides stained with Mallory’s trichrome, *n* = 5. Inflammatory process: infiltrate of leukocyte cells in the injured tissue (The infiltrate may show neutrophil, eosinophil, basophil, monocyte, and lymphocyte cells in much larger amounts than would be expected in a normal dermis). Bleeding: red blood cells in the tissue parenchyma (outside the blood vessel), confirming the presence of tissue hemorrhage. Edema: empty space in the tissue, resulting from the extravasation of aqueous fluid from the vessels into the interstitial space, which distances the cells. Neoangiogenesis: Existence of capillaries at the site of injury, which shows that vascularization is back at the site. Remodeled fibroblast: reorganization of fibroblasts at the wound site, which becomes many, suggesting that healing is taking place or has already taken place. Collagen deposit: the existence of type I collagen deposit in the injured tissue (stained in blue with Mallory’s trichromatic staining). Re-epithelialization: the creation of a new lining epithelium covering the wound site. Hair follicle: the emergence of new hair follicles (hair) in the place where before there were only characteristics of injured tissue.

## Data Availability

Not applicable.

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
