# Peer review of "Effect of Hydrogel Containing Achyrocline satureioides (Asteraceae) Extract–Loaded Nanoemulsions on Wound Healing Activity"

_pharmaceutics, 2022, doi:10.3390/pharmaceutics14122726_

Round 1

Reviewer 1 Report

Dear Editor

The submitted manuscript entitled “The effect of hydrogel-thickened nanoemulsions loaded extract of Achyrocline satureioides (Asteraceae) on wound healing activity in Wistar rats (Rattus norvegicus)” was very interesting however it needs major revision before acceptance.

·       In the title replace: Wistar rats (Rattus norvegicus) with in vivo.

·       The abstract must be contain all the obtained results referring to the corresponding methodology.  

·       Flavonoids content analysis should be evaluated with the plant extract itself.

·       Drug release and encapsulation efficiencies should investigated.

·       Skin permeation/retention studies were tried with pigskin. Why the authors did not use rats’ skin for data achievement during in vivo study.

·       Authors analysed the progression of inflammation at day 2 and 7 only however the histological studies were  evaluated during three-time intervals (2, 7 and 12). Explain

·       Figure 1 represented the wound contraction, and it was noticed that the authors analysed the data in the histogram through day 1, 3, 7 and 12 while the graphical figures denoted day 1, 2, 7 and 12 . Explain

·       Data in figure 2 showed MPO, TNF and IL-1 at only one time interval (day 2) without representing day 7 and 12

·       Figure 3 has a very poor resolution

·       Authors should use relatively recent references

·       The English language must be revised.

Author Response

Response to Reviewer 1 Comments

The submitted manuscript entitled “The effect of hydrogel-thickened nanoemulsions loaded extract of Achyrocline satureioides (Asteraceae) on wound healing activity in Wistar rats (Rattus norvegicus)” was very interesting however it needs major revision before acceptance.

Authors: We thank the reviewer for the comments, suggestions, and questions. You can find a point-by-point letter answering to the reviewer’s questions bellow. All modifications made are highlighted in red in the revised manuscript.

Point 1: In the title replace: Wistar rats (Rattus norvegicus) with in vivo.

Response 1: We sincerely appreciate the suggestion and agree that this substitution sounds better on paper.

Point 2: The abstract must be contain all the obtained results referring to the corresponding methodology. 

Response 2: We thank the reviewer for the comment. Alterations were done in manuscript’s abstract to attend the reviewer request.

Point 3: Flavonoids content analysis should be evaluated with the plant extract itself.

Response 3: We thank the reviewer for the comment. The flavonoids content analysis in plant extract was carried out according to Balestrin et al., 2020 [1].  More information was added in the new version of the manuscript.

Point 4: Drug release and encapsulation efficiencies should investigated.

Response 4: We thank the reviewer for the comment. In fact, the association efficiency was previously evaluated during the development of formulations as described by Bidone et al. (2014) [2]. The authors used ultrafiltration membranes (Ultra free-MC 10,000 MW, Merck Millipore, Billerica, USA). Aliquots of nanoemulsion were centrifugated and the ultrafiltrate obtained was used for determination of free flavonoids (quercetin, luteolin and 3-methylquercetin). No flavonoids were detected in the external aqueous phase, which indicates that, most likely due to the low water solubility of these flavonoids, they are located in the internal phase of the nanoemulsions.

Point 5:  Skin permeation/retention studies were tried with pigskin. Why the authors did not use rats’ skin for data achievement during in vivo study.

Response 5: We thank the reviewer for the question. In fact, in the present article, we carried out in vitro release and skin permeation/retention studies with the porcine ear skin, based on previously literature for skin diffusion studies, since porcine ear skin has similar histological characteristics compared to human skin [3,4]. In addition, the porcine ear skin was kindly donated from animals sacrificed at a local slaughterhouse (Ouro do Sul – Cooperativa dos Suinocultores do Caí Superior Ltda., Harmonia, BR).

Point 6: Authors analysed the progression of inflammation at day 2 and 7 only however the histological studies were evaluated during three-time intervals (2, 7 and 12). Explain

Response 6: The histological analysis was performed in order to evaluate each of the different phases of healing: inflammation, proliferation, and remodeling (Grupta). The The investigation of the parameters involved, mostly, in the inflammatory phase (MPO, IL and TNF) was performed only in the first days of treatments, because, after this period, there is a decrease in their quantification, due to the result of the healing process.

Point 7: Figure 1 represented the wound contraction, and it was noticed that the authors analysed the data in the histogram through day 1, 3, 7 and 12 while the graphical figures denoted day 1, 2, 7 and 12. Explain

Response 7: We thank the reviewer for the comment, and we apologize for the error. It was corrected and highlighted in the revised manuscript.

Point 8: Data in figure 2 showed MPO, TNF and IL-1 at only one time interval (day 2) without representing day 7 and 12

Response 8: MPO, IL1 and TNF are markers present in the inflammatory phase of the lesion. The healing phases, although not totally separate, present a cascade of events and the inflammatory phase, in its normal course of healing, occurs in the first days after the injury, so the measurement of inflammatory markers was performed only on day 02.

Point 9: Figure 3 has a very poor resolution

Response 9: We thank the reviewer for the comment. The Figure 3 was improved.

Point 10: Authors should use relatively recent references

Response 10: Thanks for the comment, recent references were added in the revised version of manuscript.

Point 11: The English language must be revised.

Response 11: We appreciate the suggestion. To attend the reviewer’s request, our manuscript was sent to a specialized English proofreading company as demonstrated by the certification append.

 References

  1.  Balestrin, L.A.; Fachel, F.N.S.; Koester, L.S.; Bassani, V.L.; Teixeira, H.F. A Stability-Indicating Ultra-Fast Liquid Chromatography Method for the Assay of the Main Flavonoids of Achyrocline Satureioides (Marcela) in Porcine Skin Layers and Nanoemulsions. Phytochemical Analysis 2020, 31, 905–914, doi:10.1002/pca.2962.
  2. Bidone, J.; Zorzi, G.K.; Carvalho, E.L.S.; Simões, C.M.O.; Koester, L.S.; Bassani, V.L.; Teixeira, H.F. Incorporation of Achyrocline Satureioides ( Lam .) DC Extracts into Topical Nanoemulsions Obtained by Means of Spontaneous Emulsification Procedure. Industrial Crops and Products 2014, 62, 421–429.
  3. Praça, F.S.G.; Medina, W.S.G.; Eloy, J.O.; Petrilli, R.; Campos, P.M.; Ascenso, A.; Bentley, M.V.L.B. Evaluation of Critical Parameters for in Vitro Skin Permeation and Penetration Studies Using Animal Skin Models. European Journal of Pharmaceutical Sciences 2018, 111, 121–132, doi:10.1016/j.ejps.2017.09.034.
  4. Sekkat, N.; Kalia, Y.N.; Guy, R.H. Biophysical Study of Porcine Ear Skin In Vitro and Its Comparison to Human Skin In Vivo. J Pharm Sci 2002, 91, 2376–2381, doi:10.1002/jps.10220.
  5. Gupta, A., Kowalczuk, M., Heaselgrave, W., Britland, S.T., Martin, C., Radecka, I., The production and application of hydrogels for Wound Management: A Review, European Polymer Journal (2018), doi: https://doi.org/10.1016/j.eurpolymj.2018.12.019

Reviewer 2 Report

This manuscript describes a hydrogel containing 73 hydroethanolic A. satureioides (Asteraceae) extract and its effect on wound healing or tissue reconstruction. After careful review, it is suggested that authors must present the results clearly. For example, in all Figures, it is not clear what is what as the figures are neither labeled nor have high quality resolution therefore it is difficult to extract the information form them.

 Authors must also clarify as to what kind of hydrogel was used. The chemical structure and schematic diagram of the hydrogel must be given. At the end of the Introduction, the authors must explain precisely about the novelty of the research by giving a schematic diagram.

The analytic data for hydrogel containing the extract is missing. How the hydrogel with/without A. satureioides was characterized?

 What about its porosity that plays a crucial role in loading and de-loading? What about its image from SEM or TEM before and after inclusion of A. satureioides molecules?

Are there any UV-Vis or fluorescence analysis for investigating the hydrogel with and without the A. satureioides? If not, it must be reasoned as to why not.

What is the sacle for Histological images (must be mentioned)?

The language of the manuscript needs to be improved as well.

Author Response

Response to Reviewer 2 Comments

This manuscript describes a hydrogel containing hydroethanolic A. satureioides (Asteraceae) extract and its effect on wound healing or tissue reconstruction. After careful review, it is suggested that authors must present the results clearly. For example, in all Figures, it is not clear what is what as the figures are neither labeled nor have high quality resolution therefore it is difficult to extract the information form them.

Authors: We thank the reviewer for the comments, suggestions, and questions. You can find a point-by-point letter answering to the reviewer’s questions bellow. All modifications made are highlighted in red in the revised manuscript. All figures were improved.

Point 1: Authors must also clarify as to what kind of hydrogel was used. The chemical structure and schematic diagram of the hydrogel must be given. At the end of the Introduction, the authors must explain precisely about the novelty of the research by giving a schematic diagram.

Response 1: We thank the reviewer for the suggestion.  Carbopol is a polymer widely used for the development of hydrogels nanotechnology-based delivery systems in views of its physicochemical properties and rheological characteristics, which are attractive for topical application. In fact, Carbopol Ultrez-21 was specifically chosen based on its faster wetting properties, higher thickening efficiency, and higher electrolyte tolerance when compared to other members of the Carbopol series [1], which could positivity impact the appearance of our final formulation. The manuscript introduction section (1.) was revised aiming to highlight the novelty of this study was defined.  

Point 2: The analytic data for hydrogel containing the extract is missing. How the hydrogel with/without A. satureioides was characterized?

Response 2: We thank the reviewer for the question. Droplet size, polydispersity index, zeta potential and flavonoids content determination were carried out to characterizes the hydrogel containing the extract and blank hydrogel. Alterations were done in manuscript’s sections 2.2.3. and 2.2.4. aiming clarify this point.

Point 3: What about its porosity that plays a crucial role in loading and de-loading? What about its image from SEM or TEM before and after inclusion of A. satureioides molecules?

Response 3: We agree with the point raised by the reviewer. In fact, the formulation used to load A. satureioides extract in this study, was previously developed, optimized and fully characterized by our research group in a previous study [2,3]. In these studies, droplet morphology through TEM micrographs of nanoemulsions before and after inclusion of A. satureioides extract incorporation was performed, and demonstrated spheroid shape, well-defined edge, and displaying a size in a 200–300 nm range with typical appearance of oil–water nanoemulsions.  The physicochemical characterization can be considered here as quality control assays for the formulation, which follows some standard tests. The results are in fact similar to those previous reported by our research group, showing the reproducibility batch-to-batch even using a complex crude plant extract, which demonstrates that experimental conditions used were well controlled.

Point 4: Are there any UV-Vis or fluorescence analysis for investigating the hydrogel with and without the A. satureioides? If not, it must be reasoned as to why not.

Response 4: We thank the reviewer for the question. In fact, the method used for determination of flavonoids content in derived hydrogels was previously developed and validated through an ultra-fast liquid chromatography (UFLC) system coupled to the detection of a photodiode array (PDA) [4]. In this study, all blank matrices (without the A. satureioides) were tested to verify the matrix effect. Overall results demonstrated a low matrix effect for all complex matric.

Point 5: What is the scale for Histological images (must be mentioned)?

Response 5:  We are sorry for the absence of these scale. We informed the scale used in legend of histological images.

Point 6: The language of the manuscript needs to be improved as well.

Response 6: We appreciate the suggestion. To attend the reviewer’s request, our manuscript was sent to a specialized English proofreading company as demonstrated by the certification append.

References

  1. Lubrizol Advanced Materials, Inc. Carbopol® Ultrez 21 Polymer Technical Data Sheet. 2002, 1–4.
  2. Bidone, J.; Zorzi, G.K.; Carvalho, E.L.S.; Simões, C.M.O.; Koester, L.S.; Bassani, V.L.; Teixeira, H.F. Incorporation of Achyrocline Satureioides (Lam.) DC Extracts into Topical Nanoemulsions Obtained by Means of Spontaneous Emulsification Procedure. Ind Crops Prod 2014, 62, 421–429, doi:10.1016/j.indcrop.2014.08.042.
  3. Balestrin, L.A.; Kreutz, T.; Fachel, F.N.S.; Bidone, J.; Gelsleichter, N.E.; Koester, L.S.; Bassani, V.L.; Braganhol, E.; Dora, C.L.; Teixeira, H.F. Achyrocline Satureioides (Lam.) Dc (Asteraceae) Extract-Loaded Nanoemulsions as a Promising Topical Wound Healing Delivery System: In Vitro Assessments in Human Keratinocytes (Hacat) and Het-Cam Irritant Potential. Pharmaceutics 2021, 13, doi:10.3390/pharmaceutics13081241.
  4. Balestrin, L.A.; Fachel, F.N.S.; Koester, L.S.; Bassani, V.L.; Teixeira, H.F. A Stability-Indicating Ultra-Fast Liquid Chromatography Method for the Assay of the Main Flavonoids of Achyrocline Satureioides (Marcela) in Porcine Skin Layers and Nanoemulsions. Phytochemical Analysis 2020, 31, 905–914, doi:10.1002/pca.2962.

Reviewer 3 Report

The scientific quality of the manuscript is insufficient for publication in its current form. Specific questions and points requiring attention are itemized below.

1.      Reviewer comments: The authors must improve the Abstract. For example, the authors describe that “Recent studies have demonstrated the effect of Achyrocline satureioides (Asteraceae) extracts on keratinocytes and fibroblasts proliferation and collagen renovation. Such activities have been related to the flavonoid aglycones content in the extracts”. This paragraph is similar to an introduction section…

 2.      Reviewer comments: It is still difficult to find the novelty of the work concerning what has already been published. What is the difference between what is published with what the authors want to publish? It is not clear. The authors must describe these differences in the introduction section.

 3.      Reviewer comments: The study is related to the nanoencapsulation of extract put into a hydrogel. However, in the introduction sections, information and antecedents about the nanoemulsions system and hydrogels are not described.

 4.      Reviewer comments: Why the authors used nanoemulsions systems and hydrogel? It is not clear.

5.      Reviewer’s comment: The results and discussion sections are poor. More comparisons with previous literature should be discussed.

6.      Reviewer comments: The authors must indicate if significant differences exist in the results obtained in all Tables and Figures. The authors can use letters or numbers  (p<0.05).

7.      Reviewer comments: what is the importance of evaluating the droplet size, PI, and zeta potential on emulsions?

8.      Reviewer comments: The objective of the manuscript is related to the production of nanoemulsions. However, the mean droplet size of the emulsions is 250 nm. What droplet size of emulsions is considered nano?????? The results obtained did not demonstrate that nanoemulsions were obtained, so the title must be modified.

9.      Reviewer comments: The authors describe that “A higher negative zeta potential was observed for the formulation containing the extract of A. satureioides, suggesting the adsorption of extract components at the oil/water interface of nanoemulsions, such as organic acids, as already reported in the literature”. This assertion is weak. Why the emulsion displayed a negative charge??? What is the charge attributed to? Are the values good??? Bad?? Explain more in detail.

10.  Reviewer comments. Explain and discuss the references put in the Results and discussion section. Don't just mention them.

11.  Reviewer comments. The authors must evaluate antioxidant capacity. For example, authors must determine DPPH and ABTS.

12.  Reviewer comments: The authors must determine the main compounds of Achyrocline satureioides extract.

13.  Reviewer comments: How do the authors demostrate that the hydrogel was formed? The authors must display images of hydrogels produced and microscopic images of emulsions.

14. Reviewer comments: What properties of hydrogels were evaluated?. 

15. Reviewer comments: Line 347: "The interesting results that we observed with A. satureioides may be related to the content of phenolic compounds resulting from the differences in the extraction conditions (ethanol content and maceration time), as well as with the formulation used for the administration of the extract from this medicinal plant". In this study, the authors did not evaluate the extraction conditions. How do the authors assert this claim?

16. Reviewer comments: Explain in detail each parameter evaluated in Table 5. 

Author Response

Response to Reviewer 3 Comments

The scientific quality of the manuscript is insufficient for publication in its current form. Specific questions and points requiring attention are itemized below.

Authors: We thank the reviewer for the comments, suggestions, and questions. You can find a point-by-point letter answering to the reviewer’s questions bellow. All modifications made are highlighted in red in the revised manuscript.

Point 1.      Reviewer comments: The authors must improve the Abstract. For example, the authors describe that “Recent studies have demonstrated the effect of Achyrocline satureioides (Asteraceae) extracts on keratinocytes and fibroblasts proliferation and collagen renovation. Such activities have been related to the flavonoid aglycones content in the extracts”. This paragraph is similar to an introduction section…

Response 1: We thank the reviewer for the comment. Alterations were done in manuscript’s abstract to attend the reviewer request.

Point 2.     Reviewer comments: It is still difficult to find the novelty of the work concerning what has already been published. What is the difference between what is published with what the authors want to publish? It is not clear. The authors must describe these differences in the introduction section.

Response 2: Thank you for your suggestion. We certainly agree with the reviewer. In fact, in the present article, we carried out for the first time the in vivo evaluation of Achyrocline satureioides extract-loaded nanoemulsions thickened in Carbopol hydrogels activity in topical wound healing in Wistar rats. The manuscript introduction section (1.) was revised aiming to highlight the novelty of this study was defined.

Point 3.      Reviewer comments: The study is related to the nanoencapsulation of extract put into a hydrogel. However, in the introduction sections, information and antecedents about the nanoemulsions system and hydrogels are not described.

Response 3: We thank the reviewer for the comment. Alterations were done in manuscript to attend the reviewer request.

Point 4.      Reviewer comments: Why the authors used nanoemulsions systems and hydrogel? It is not clear.

Response 4: We thank the reviewer for the question. Our research group has recently been interested in the development of topical nanoemulsions as delivery systems for Achyrocline satureioides ethanolic extract [1–3]. Nevertheless, for topical application, the low viscosity from nanoemulsions is a concern and the design of a semisolid formulation is a key consideration. In the literature, this is the first report that described the development of semisolid formulations containing Achyrocline satureioides L. extract for topical application in wound healing and use nanotechnology- based products for the improvement of flavonoids skin retention/ permeation, since their small droplet sizes can improve active lipophilic molecules penetration through the skin, increasing their topical effect.

Point 5. Reviewer’s comment: The results and discussion sections are poor. More comparisons with previous literature should be discussed.

Response 5: We thank the reviewer for the comment. Alterations were done in manuscript to attend the reviewer request.

Point 6.      Reviewer comments: The authors must indicate if significant differences exist in the results obtained in all Tables and Figures. The authors can use letters or numbers (p<0.05).

Response 6: We thank the reviewer for the comment. We added the significant differences in all tables and figures.

Point 7.      Reviewer comments: what is the importance of evaluating the droplet size, PI, and zeta potential on emulsions?

Response 7: We thank the reviewer for the question. In fact, droplet size, PDI and zeta potential on emulsions can directly affects the physical stability of these dispersed systems. Reduced droplet size allows for more droplets to be distributed on the interface, resulting in a more stable emulsion. In addition, higher zeta potential values are crucial factor to determining desirable physical stability of the nanoemulsions.

Point 8.      Reviewer comments: The objective of the manuscript is related to the production of nanoemulsions. However, the mean droplet size of the emulsions is 250 nm. What droplet size of emulsions is considered nano?????? The results obtained did not demonstrate that nanoemulsions were obtained, so the title must be modified.

Response 8:  We thank the reviewer for the question. In drug delivery systems, nanoemulsions are dispersions of two immiscible liquids (oil-in-water) stabilized by an appropriate surfactant system, that have been widely studied as a promising alternative for pharmaceutical and cosmetical purposes in our research group [1,4–6]. The mean droplet diameter attained is usually <500 nm [7].

Point 9.      Reviewer comments: The authors describe that “A higher negative zeta potential was observed for the formulation containing the extract of A. satureioides, suggesting the adsorption of extract components at the oil/water interface of nanoemulsions, such as organic acids, as already reported in the literature”. This assertion is weak. Why the emulsion displayed a negative charge??? What is the charge attributed to? Are the values good??? Bad?? Explain more in detail.

Response 9: We thank the reviewer for the question. The nanoemulsions exhibited a negative value of zeta potential related with the presence of negatively charged phospholipids and free fatty acids present in egg-lecithin. A higher negative zeta potential was observed for the formulation containing the extract of A. satureioides and may be a consequence of the negatively charged phospholipids and free fatty acids in the egg yolk lecithin and the extract components located at the o/w interface of the nanoemulsions, as phenolic acids, as previously demonstrated by our research group [1,3]. Alterations were done in manuscript aiming clarify this point.

Point 10.  Reviewer comments. Explain and discuss the references put in the Results and discussion section. Don't just mention them.

Response 10: We thank the reviewer for the comment. Alterations were done in manuscript to attend the reviewer request.

Point 11.  Reviewer comments. The authors must evaluate antioxidant capacity. For example, authors must determine DPPH and ABTS.

Response 11: We agree with the point raised by the reviewer. In fact, the formulation used to load A. satureioides extract in this study, was previously studied by our research group [8] in order to evaluate its antioxidant capacity.

Point 12.  Reviewer comments: The authors must determine the main compounds of Achyrocline satureioides extract.

Response 12: We thank the reviewer for the comment. The flavonoids content analysis in plant extract was carried out according to Balestrin et al., 2020 [9].  More information was added in the new version of the manuscript.

Point 13.  Reviewer comments: How do the authors demonstrate that the hydrogel was formed? The authors must display images of hydrogels produced and microscopic images of emulsions.

Response 13: We agree with the point raised by the reviewer. In fact, the formulation used to load A. satureioides extract in this study, was previously developed, optimized and fully characterized by our research group in a previous study [2,3]. In these studies, droplet morphology through TEM micrographs of nanoemulsions before and after inclusion of A. satureioides extract incorporation was performed, and demonstrated spheroid shape, well-defined edge, and displaying a size in a 200–300 nm range with typical appearance of oil–water nanoemulsions.  The physicochemical characterization can be considered here as quality control assays for the formulation, which follows some standard tests. The results are in fact similar to those previous reported by our research group, showing the reproducibility batch-to-batch even using a complex crude plant extract, which demonstrates that experimental conditions used were well controlled.

Point 14. Reviewer comments: What properties of hydrogels were evaluated?

Response 14: We thank the reviewer for the question. Droplet size, polydispersity index, zeta potential and flavonoids content determination were carried out to characterizes the hydrogel containing the extract and blank hydrogel. Alterations were done in manuscript’s sections 2.2.3. and 2.2.4. aiming clarify this point.

Point 15. Reviewer comments: Line 347: "The interesting results that we observed with A. satureioides may be related to the content of phenolic compounds resulting from the differences in the extraction conditions (ethanol content and maceration time), as well as with the formulation used for the administration of the extract from this medicinal plant". In this study, the authors did not evaluate the extraction conditions. How do the authors assert this claim?

Response 15: The set of results obtained in our study showed promising results for A. satureioides. We relate these good results with the concentration of phenolic compounds in the extract. In our case, the concentration of phenolic compounds is different from the study by Pereira et al. [10], which may be related to the extraction conditions. Our results must also have been influenced by the developed nanostructured formulation.

Point 16. Reviewer comments: Explain in detail each parameter evaluated in Table 5.

Response 16: We appreciate the reviewer's suggestion. Below is a brief description of each parameter evaluated in Table 5: Qualitative histological analysis was performed on slides stained with Mallory's trichrome.

Inflammatory process: infiltrate of leukocyte cells in the injured tissue (The infiltrate may show neutrophil, eosinophil, basophil, monocyte, and lymphocyte cells in much larger amounts than would be expected in a normal dermis).

Bleeding: red blood cells in the tissue parenchyma (outside the blood vessel), confirming the presence of tissue hemorrhage.

Edema: empty space in the tissue, resulting from the extravasation of aqueous fluid from the vessels into the interstitial space, which distances the cells.

Neoangiogenesis: Existence of capillaries at the site of injury, which shows that vascularization is back at the site.

Remodeled fibroblast: reorganization of fibroblasts at the wound site, which becomes many, suggesting that healing is taking place or has already taken place.

Collagen deposit: the existence of type I collagen deposit in the injured tissue (stained in blue with Mallory's trichromatic staining).

Re-epithelialization: the creation of a new lining epithelium covering the wound site.

Hair follicle: the emergence of new hair follicles (hair) in the place where before there were only characteristics of injured tissue.

 References

  1.  Bidone, J.; Zorzi, G.K.; Carvalho, E.L.S.; Simões, C.M.O.; Koester, L.S.; Bassani, V.L.; Teixeira, H.F. Incorporation of Achyrocline Satureioides (Lam.) DC Extracts into Topical Nanoemulsions Obtained by Means of Spontaneous Emulsification Procedure. Ind Crops Prod 2014, 62, 421–429, doi:10.1016/j.indcrop.2014.08.042.
  2. Balestrin, L.A.; Bidone, J.; Bortolin, R.C.; Moresco, K.; Moreira, J.C.; Teixeira, H.F. Protective Effect of a Hydrogel Containing Achyrocline Satureioides Extract-Loaded Nanoemulsion against UV-Induced Skin Damage. J Photochem Photobiol B 2016, 163, 269–276, doi:10.1016/j.jphotobiol.2016.08.039.
  3. Balestrin, L.A.; Kreutz, T.; Fachel, F.N.S.; Bidone, J.; Gelsleichter, N.E.; Koester, L.S.; Bassani, V.L.; Braganhol, E.; Dora, C.L.; Teixeira, H.F. Achyrocline Satureioides (Lam.) Dc (Asteraceae) Extract-Loaded Nanoemulsions as a Promising Topical Wound Healing Delivery System: In Vitro Assessments in Human Keratinocytes (Hacat) and Het-Cam Irritant Potential. Pharmaceutics 2021, 13, doi:10.3390/pharmaceutics13081241.
  4. Teixeira, H.F.; Bruxel, F.; Fraga, M.; Schuh, R.S.; Zorzi, G.K.; Matte, U.; Fattal, E. Cationic Nanoemulsions as Nucleic Acids Delivery Systems. Int J Pharm 2017, 534, 356–367, doi:10.1016/j.ijpharm.2017.10.030.
  5. Fasolo, D.; Bassani, V.L.; Teixeira, H.F. Development of Topical Nanoemulsions Containing Quercetin and 3- O -Methylquercetin. Pharmazie 2009, 64, 726–730, doi:10.1691/ph.2009.8357.
  6. Fachel, F.N.S.; Medeiros-Neves, B.; Dal Prá, M.; Schuh, R.S.; Veras, K.S.; Bassani, V.L.; Koester, L.S.; Henriques, A.T.; Braganhol, E.; Teixeira, H.F. Box-Behnken Design Optimization of Mucoadhesive Chitosan-Coated Nanoemulsions for Rosmarinic Acid Nasal Delivery—In Vitro Studies. Carbohydr Polym 2018, 199, 572–582, doi:10.1016/j.carbpol.2018.07.054.
  7. Singh, Y.; Meher, J.G.; Raval, K.; Khan, F.A.; Chaurasia, M.; Jain, N.K.; Chourasia, M.K. Nanoemulsion: Concepts, Development and Applications in Drug Delivery. Journal of Controlled Release 2017, 252, 28–49.
  8. Zorzi, G.K.; Caregnato, F.; Moreira, J.C.F.; Teixeira, H.F.; Carvalho, E.L.S. Antioxidant Effect of Nanoemulsions Containing Extract of Achyrocline Satureioides (Lam) D.C.—Asteraceae. AAPS PharmSciTech 2016, 17, 844–850, doi:10.1208/s12249-015-0408-8.
  9. Balestrin, L.A.; Fachel, F.N.S.; Koester, L.S.; Bassani, V.L.; Teixeira, H.F. A Stability-Indicating Ultra-Fast Liquid Chromatography Method for the Assay of the Main Flavonoids of Achyrocline Satureioides (Marcela) in Porcine Skin Layers and Nanoemulsions. Phytochemical Analysis 2020, 31, 905–914, doi:10.1002/pca.2962.
  10. Pereira, L.X.; Silva, H.K.C.; Longatti, T.R.; Silva, P.P.; di Lorenzo Oliveira, C.; de Freitas Carneiro Proietti, A.B.; Thomé, R.G.; Vieira, M. do C.; Carollo, C.A.; Demarque, D.P.; et al. Achyrocline Alata Potentiates Repair of Skin Full Thickness Excision in Mice. J Tissue Viability 2017, 26, 289–299, doi:10.1016/j.jtv.2017.09.005.

Round 2

Reviewer 1 Report

The submitted manuscript entitled “The Effect of Hydrogel Containing Achyrocline satureioides 2 (Asteraceae) Extract–Loaded Nanoemulsions on Wound Healing 3 Activity In Vivo” was very interesting however only some English and typing mistakes were noticed 

Author Response

Response to Reviewer 1 Comments

The submitted manuscript entitled “The Effect of Hydrogel Containing Achyrocline satureioides (Asteraceae) Extract–Loaded Nanoemulsions on Wound Healing Activity In Vivo” was very interesting however only some English and typing mistakes were noticed.

Authors: We thank the reviewer for the comments, suggestions, and questions. Our manuscript was sent to a specialized English proofreading company as demonstrated by the certification append.

Reviewer 2 Report

Title is not accurate. Delete 'The' from it. It can be rewritten as " Effect of Hydrogel Containing Achyrocline satureioides 2 (Asteraceae) Extract–Loaded Nanoemulsions on Wound Healing Activity". 

Also, authors must provide a chemical structure in hydrogel and a representative structure of the loaded compound or drug. It can be included in a Scheme (diagrammatic scheme) at the end of 'Introduction' section. 

Author Response

Response to Reviewer 2 Comments

Title is not accurate. Delete 'The' from it. It can be rewritten as " Effect of Hydrogel Containing Achyrocline satureioides (Asteraceae) Extract–Loaded Nanoemulsions on Wound Healing Activity".

Authors: We sincerely appreciate the suggestion and agree that this substitution sounds better on paper.

Also, authors must provide a chemical structure in hydrogel and a representative structure of the loaded compound or drug. It can be included in a Scheme (diagrammatic scheme) at the end of 'Introduction' section.

Authors: We thank the reviewer for the suggestions. We included a scheme at the end of Introduction’s section. All modifications made are highlighted in red in the revised manuscript.

Reviewer 3 Report

The article can be accepted once the following are met:

Reviewers comments:: The authors must indicate if significant differences exist in the results obtained in all Tables. The authors can use letters or numbers (p<0.05).

Author Response

Response to Reviewer 3 Comments

The article can be accepted once the following are met:

Reviewers’ comments: The authors must indicate if significant differences exist in the results obtained in all Tables. The authors can use letters or numbers (p<0.05).

Authors: We thank the reviewer for the comments. We added the significant differences in all tables. All modifications made are highlighted in red in the revised manuscript.